# The Effect of Drenching (Very) Low Birth Weight Piglets with a Dense, Concentrated Milk Replacer at Farms with Differing Farrowing Management

**DOI:** 10.3390/ani13010063

**Published:** 2022-12-23

**Authors:** Kevin Van Tichelen, Sara Prims, Miriam Ayuso, Lieselotte Van Bockstal, Céline Van Kerschaver, Mario Vandaele, Jeroen Degroote, Steven Van Cruchten, Joris Michiels, Chris Van Ginneken

**Affiliations:** 1Comparative Perinatal Development, Faculty of Biomedical, Pharmaceutical and Veterinary Sciences, Antwerp University, Universiteitsplein 1, 2610 Wilrijk, Belgium; 2Laboratory for Animal Production and Animal Product Quality, Faculty of Bioscience Engineering, Ghent University, Coupure Links 653, 9000 Ghent, Belgium

**Keywords:** pig, performance, oral supplementation, neonatal, survival, low birthweight, milk replacer, neonatal management, perinatal management

## Abstract

**Simple Summary:**

Over the past three decades, sows’ litter sizes have been increased to improve productivity, but have also led to increased proportions of (very) low birth weight piglets, and consequently, higher pre-weaning mortality. One possible intervention to counter the increased mortality is supplementing a milk replacer. This study aimed to determine if the performance of low birth weight piglets can be improved by drenching a dense, concentrated milk replacer and whether the frequency of drenching and the severity of the low birth weight played a role. Secondly, this study compared the supplementation of the same milk replacer at two farms with different perinatal management. No effect of drenching a dense milk replacer on the survival or performance of (very) low birth weight piglets was observed, regardless of farm and, apparently, of the applied management. However, mortality rates were lower at the farm with a higher level of perinatal management, suggesting that high-quality care might have more effect on the survival of small piglets than drenching a dense milk replacer.

**Abstract:**

Introducing hyperprolific sows has led to proportionally more (very) low birth weight ((V)LBW) piglets, accompanied by higher mortality. To improve the survival of (V)LBW piglets, drenching a dense milk replacer (DMR) could be applied. A first experiment evaluated the effect of drenching DMR (1 or 3 doses within 24 h after birth) to LBW ((mean litter birth weight − 1*SD) and weighing between 1 kg and 750 g) and VLBW piglets ((mean litter birth weight − 1.5*SD) and weighing less than 750 g). On days 1, 2, 3, 9, and two days post-weaning, body weight, growth, skin lesions, and mortality were monitored. No effect of DMR was observed on any of the parameters. In a second experiment, LBW piglets were supplemented with DMR (similarly to experiment 1) at two farms differing in the level of perinatal care. The same parameters were evaluated, and again none were affected by drenching DMR. Overall survival of the LBW piglets was significantly higher at the farm with high perinatal care. It can be concluded that good perinatal management is more effective in enhancing the survival of LBW piglets than drenching.

## 1. Introduction

To improve the efficiency of sow husbandry, there has been a strong selection over the past three decades for highly prolific sows. This genetic selection has resulted in higher litter sizes and, from an economic point of view, improved productivity [1,2]. However, this increase in litter size is accompanied by a higher variation in birth weight and a larger proportion of piglets with an abnormally low birth weight [3,4]. These low birth weight (LBW) piglets are often unable to ingest adequate amounts of colostrum due to reduced vitality and an impaired ability to compete with larger littermates for a functional teat [5,6]. Consequently, LBW piglets are more susceptible to starvation, hypothermia, and crushing [7,8,9,10]. Increased litter sizes have thus not necessarily led to increased profits, as they are often accompanied by elevated pre-weaning morbidity and mortality. This raises animal welfare and ethical concerns from the public for both sows and piglets, and challenges pig producers to keep LBW piglets alive while minimizing any negative impact on their production numbers [11,12,13]. A management strategy that aims to improve the survival and performance of LBW piglets, and one that has been the subject of several studies, is manually providing neonatal piglets with a nutritional supplement via drenching. Different compounds, such as colostrum, prebiotics, antioxidants, or energy boosters, can be drenched [14,15,16,17,18,19,20,21,22]. However, the lack of a consistent across-studies criterion to identify the targeted LBW piglets limits the comparison of results between studies and obscures drawing a conclusion on the effect of these compounds (reviewed by [23,24,25]). While some authors apply birth weight cut-offs up to 1.00–1.35 kg to define LBW piglets [14,17,18,19], others use lower birth weights [15,26,27,28]. Moreover, some studies even consider piglets merely below one kilogram as very low birth weight piglets [18] while other studies consider piglets in this weight range as LBW piglets [15,17,19,26,27]. Thus, it remains to be determined in what conditions (e.g., birth weight, perinatal management) and which compounds exert a beneficial effect on piglet performance. Moreover, in the case that drenching would significantly improve piglet performance, additional factors should be taken into account before drenching can be advised as a pre-weaning strategy, i.e., the labour costs and intensity (individual and/or repeated handling of supplemented piglets), and financial costs (supplemented products, spillage) associated with the nutritional support via drenching [24].

The present study was designed to evaluate the supplementation of a dense, concentrated milk replacer on the survival and performance of LBW piglets.

The first experiment examined whether drenching a dense milk replacer affected the performance (growth, survival) of low birth weight piglets and what frequency of drenching was optimal. It was hypothesized that the milk replacer would improve their performance indirectly by supplying enough energy so the piglet can achieve a (first) suckle. Additionally, it was tested whether this boosting effect would be higher when the milk replacer was drenched three times versus only once. Simultaneously the existence of a lower limit—in terms of birth weight—up to which drenching would have an effect was assessed. It was hypothesized that piglets with a birth weight below 750 g were too weak to experience any benefits from drenching a milk replacer.

In a second step, the experiment was repeated at another farm. This allows us to check the reproducibility of the results at the first farm and to assess the impact of possible farm-specific conditions. In this experiment both farms differed in their perinatal management (partus induction, neonatal supervision, heat provision). It was hypothesized that, in the case of a higher level of perinatal care, drenching a dense milk replacer would improve the performance of LBW piglets more than in the case of lower perinatal care.

## 2. Materials and Methods

### 2.1. Ethical Approval

This study was reviewed and approved by the Ethical Committee for Animal Experimentation of the University of Antwerp (ECD 11/2018) and was compliant with national legislation and European guidelines (2010/63/EC).

### 2.2. Farms and Animals

The study consisted of two consecutive field trials that were conducted on a commercial farm in Meer (Farm A, Hoogstraten, Belgium) and Loenhout (Farm B, Wuustwezel, Belgium). The main animal and management traits of the two farms are presented in Table 1.

All piglets included in the study, as well as their littermates, were subjected to the standard handling procedures in the farm: before the age of one week, all piglets were ear-tagged, tail-docked, received a 200 mg iron dextran intramuscular injection, and all male piglets were castrated using meloxicam analgesics. Sows that had been used for piglet selection during previous farrowing rounds were not considered in later farrowing rounds, thus each sow was only included once. Each of the treatment groups comprised equal numbers of sex and birth weight category piglets.

### 2.3. Piglet Selection

#### Experiment 1: Farm A

All piglets were weighed within four hours after birth. When parturition was finished, the mean birth weight of each litter (mean BW_litter_) and the SD were calculated. Additionally, the mean birth weight of all piglets that were born from 139 sows (BW_population_, *n* = 2337) and the SD were calculated. When considering BW_population_ − 1 SD, the resulting birth weight was less than 1 kg. When considering BW_population_ − 1.5 SD, the birth weight was less than 750 g. Subsequently, LBW and VLBW piglets were selected using two criteria, based on both the farm population’s and the litter’s mean birth weight:

Piglets with a birth weight between (mean BW_litter_ − 1 SD) and (mean BW_litter_ − 1.5 SD) and weighing between 750 g and 1 kg were categorised as LBW piglets.

Piglets with a birth weight between (mean BW_litter_ − 1.5 SD) and (mean BW_litter_ − 2.5 SD) and weighing less than 750 g were categorised as VLBW piglets.

In total, 80 LBW and 80 VLBW piglets were selected during seven farrowing rounds from 92 sows (Table 1). An equal number of male and female piglets was selected.

#### Experiment 2: Farm B

Following the poor survival of VLBW piglets at Farm A (see results section), only LBW piglets were selected at Farm B (*n* = 150) (Table 1), spread over eight farrowing rounds and 98 sows and having an equal number of male and female piglets. Like Farm A, the mean birth weight of all piglets that were born from 230 sows was calculated (BW_population_, *n* = 4473) which—after deducting 1 SD and 1.5 SD—resulted in birth weights less than 1 kg and 750 g, respectively. Thus, the same selection criteria for LBW piglets that were used at Farm A were applied during the selection at Farm B.

### 2.4. Experimental Treatments

All piglets were randomly allocated to one of five different treatments (by using ear tags) after they were categorised as LBW or VLBW piglets in the case of Farm A and LBW piglets only in the case of Farm B.

The first treatment group received a single oral supplementation of a dense milk replacer (DMR) providing nutrients and energy in a low volume (5 mL). The DMR was prepared by dissolving 6 g of a plain milk replacer (Piglet Milk R714^®^, Table 2) in 4 mL of water (40 °C) at the time of drenching. Every 5 mL dose of milk replacer provided 59.61 kJ metabolizable energy to the drenched piglet. This group of piglets was drenched with 5 mL of DMR only once and immediately after being ear-tagged. A second treatment group was drenched three times with DMR: immediately after ear-tagging on day 1, in the evening of day 1, and in the morning of day 2. Two sham treatments were used as control groups. In the first group, animals were sham-drenched once by holding an empty 5 mL syringe in the mouth for 20 s (duration of drenching), immediately after ear-tagging. In a second sham group, the piglets were also sham drenched, but three times (morning day 1, evening day 1, and morning day 2). After (sham) drenching, each piglet was returned to the litter in a standardised way: every piglet was returned to a teat or, if the sow was not lying down on her side, amongst its siblings. To ensure any observed effect could be attributed to the product rather than the act of drenching, and to evaluate any direct influence of handling during drenching, a negative control group was added in which the animals were not drenched and only handled during data collection. This was a fifth treatment group. Within each treatment group, the number of female and male piglets (and birth weight category in the case of Farm A) was equal. The litter sizes were not standardised in size. Moreover, since the treatments were allocated on the piglet level, a single sow could have 1 up to 5 piglets belonging to the different experimental groups. The same treatments were used during both experiments.

### 2.5. Data Collection

#### 2.5.1. Body Weight and Growth

All piglets were weighed on the day of birth (day 1), day 2, day 3, day 9, and two days after weaning (day 24 and day 26 at farms A and B, respectively). All animals were weighed before drenching. Average daily growth (ADG) was calculated from day 1 until two days post-weaning, using the following formula: (body weight_day x_ − birth weight)/(x − 1). Additionally, to determine growth relative to the individual birth weight, the factorial growth (body weight_day x_/birth weight), metabolic weight (body weight^0.75^), and factorial metabolic rate (metabolic weight_day x_/metabolic weight_day 1_) were calculated.

#### 2.5.2. Colostrum Intake

Drenching DMR could result in an energy boost that allows piglets to obtain their first suckle faster, or by contrast, result in a filled stomach and a satiated feeling, resulting in a delayed first suckle. Thus, DMR could affect colostrum intake and given the importance of colostrum for the performance of the piglet [29,30], affect their subsequent performance. To determine whether supplementing 5 mL of DMR would have any of the effects, the colostrum intake (CI) was determined whenever the exact time of birth and the body weight immediately after birth were known of both the firstborn piglet and the (V)LBW piglet. This resulted in the estimation of CI of a subset of VLBW (33 piglets, 18 females, 15 males) and LBW (Farm A: 38 piglets, 17 females, 21 males; Farm B: 33 piglets, 21 females, 12 males) piglets. To calculate the CI, the mechanistic model as described by Theil et al. [31] was applied, using the weight gain after 24 h (WG, g), the body weight at birth (BWB, kg), and the suckling duration from birth until 24 h after the birth of the litter’s first piglet (D, min):CI = −106 + (2.26 × WG) + (200 × BWB) + (0.111 × D) − (1414 × (WG/D)) + (0.0182 × (WG/BWB))(1)

#### 2.5.3. Skin Lesion Scoring

A skin lesion score (for the entire body) was given using the scoring system according to Rundgren and Löfquist [32], Pluske and Williams [33], and Parrat et al. [34]:

0: no lesions

1: <5 superficial lesions (skin unbroken)

2: 5–10 superficial lesions or <5 deep lesions (skin broken and evidence of haemorrhage)

3: >10 superficial lesions or >5 deep lesions

The skin lesion scoring was performed on day 1, day 2, day 3, day 9, and two days post-weaning.

#### 2.5.4. Mortality

The number of piglets that died was inventoried 24 h after birth (day 2), on day 3, day 9, and two days after weaning. An animal was registered as deceased (day of death) on the day it was no longer observed (discarded by the farmer). Given the fixed time point at which the animals were checked, mortality could only be registered by the absence of a piglet and not by the cause of death.

#### 2.5.5. Cortisol and Chromogranin A

At Farm B, saliva was sampled from 30 LBW piglets at the age of 47 days to determine whether sham drenching would induce an acute stress response, and consequently, have an impact on any potential effect of the DMR supplementation (results presented in Appendix A (Appendix A [35,36,37,38]).

### 2.6. Statistical Analysis

To meet normality and/or homoscedasticity, body weight, factorial growth, metabolic weight, and factorial metabolic rate were log-transformed, while the other outcome variables required no transformations. Effects were considered statistically significant if *p* ≤ 0.05. *Post hoc* analysis with Tukey’s correction was used to compare different groups. All values are presented as median ± SD.

#### Experiment 1: Farm A

To evaluate the potential effect of drenching DMR on body weight, growth and colostrum intake, linear mixed models were fitted in JMP Pro 15.2 (SAS Institute Inc., Cary, NC, USA). Treatment, age, sex, and birth weight category (LBW or VLBW) were included as fixed effects. In addition, all relevant interactions between treatment, age, birth weight category, and sex were added to the model. Given that the piglets were selected over several farrowing rounds, farrowing round was added as a random effect. To account for the dependence between littermates or between measurements on the same piglet (but at a different age), random factors for sow (nested in the farrowing round) and piglet (nested in sow) were included. This starting model was simplified using stepwise backward modelling, during which all non-significant effects were removed from the starting model.

To create a ranking for the probability of more severe skin lesions occurring in certain groups, an ordinal logistic regression model was used in which treatment, age, sex, birth weight category, and their interactions were considered model effects. This model was simplified using stepwise backward modelling by only retaining significant (*p* ≤ 0.05) effects.

The probability of higher mortality between the different groups was evaluated by Cox’s proportional hazard model. Treatment, age, sex, birth weight category, and their interactions were added as fixed factors. This model was simplified using stepwise backward modelling (*p* ≤ 0.05). A *post hoc* analysis was performed using risk ratios. Additionally, mortality was visualized using Kaplan-Meier curves.

#### Experiment 2: Farm B

To analyse the potential effect of drenching DMR on body weight, growth and colostrum intake, linear mixed models were fitted in JMP Pro 15.2 (SAS Institute Inc., Cary, NC, USA). In this second experiment we only drenched LBW piglets. As a result, treatment, age, and sex, were included as fixed effects. In addition, all relevant interactions between these fixed factors were added to the model. Given that the piglets were selected over several farrowing rounds and, to account for the dependence between littermates and between measurements on the same piglet (but at a different age), random factors for farrowing round, sow (nested in the farrowing round) and piglet (nested in sow) were included. This starting model was simplified using stepwise backward modelling, during which all non-significant effects (*p* > 0.05) were removed from the starting model.

To create a ranking for the probability of more severe skin lesions occurring in certain groups, an ordinal logistic regression model was used in which treatment, age, sex, and their interactions were considered model effects. This model was simplified using stepwise backward modelling by only retaining significant (*p* ≤ 0.05) effects.

The probability of higher mortality between the different groups was evaluated by Cox’s proportional hazard model. Treatment, age, sex, and their interactions were added as fixed factors. This model was simplified using stepwise backward modelling using the stepwise backward method (*p* ≤ 0.05). A *post hoc* analysis was performed using risk ratios. Additionally, mortality was visualized using Kaplan-Meier curves.

#### Farm A vs. Farm B

In a second step, the data of both farms were combined in one large dataset to evaluate any differences between the two farms in the potential effect of drenching DMR. For possible differences in view of body weight and colostrum intake, linear mixed models were fitted in JMP Pro 15.2 (SAS Institute Inc., Cary, NC, USA) where farm was included as a fixed effect (next to treatment, age, and sex). In addition, all relevant interactions between farm, and the other fixed factors were included. Similar to the other analyses, farrowing round, sow, and piglet were added as random effects. This starting model was simplified using stepwise backward modelling, by removing those factors that had non-significant effects (*p* > 0.05) from the starting model.

The probability of more severe skin lesions in one of the farms was assessed using an ordinal logistic regression model with farm, treatment, age, sex, and their interactions considered model effects. This model was analysed using stepwise backward modelling by only retaining significant (*p* ≤ 0.05) effects.

The probability of higher mortality in one of the farms was evaluated by Cox’s proportional hazard model. Farm, treatment, age, sex, and their interactions were added as fixed factors using the stepwise backward method (*p* ≤ 0.05). A *post hoc* analysis was performed using risk ratios. Additionally, mortality was visualized using Kaplan-Meier curves.

## 3. Results

### 3.1. Experiment 1

#### 3.1.1. Body Weight and Growth

None of the fixed factors (treatment, age, sex, and birth weight category) showed a significant interaction for the different parameters related to body weight and growth. These interactions were removed as fixed factors from the linear mixed model, retaining only the individual fixed factors: treatment, age, sex, and birth weight. There was no effect of drenching DMR on the body weight (*p* = 0.179) (Figure 1), ADG (*p* = 0.091), factorial growth (*p* = 0.130), metabolic weight (*p* = 0.430) or factorial metabolic rate (*p* = 0.149). As expected, body weight increased during the experimental period (*p* < 0.001), as did factorial growth (*p* < 0.001), metabolic weight (*p* < 0.001), and factorial metabolic rate (*p* < 0.001). Understandably, the body weight of VLBW piglets was lower than that of LBW piglets (*p* < 0.001). It remained lower throughout the investigated period (since there was no significant interaction between birth weight category and age) (Figure 1). Consequently, the VLBW piglets also had a significantly lower ADG (*p* = 0.022) and metabolic weight (*p* < 0.001) than LBW piglets, and this throughout the study period. However, the factorial growth and factorial metabolic rate did not differ between the two birth weight categories (*p* = 0.683 and *p* = 0.685, respectively) (Table 3). No sex effect was observed for body weight: *p* = 0.595, ADG: *p* = 0.527, factorial growth: *p* = 0.452, metabolic weight: *p* = 0.709, and factorial metabolic rate: *p* = 0.452 (Appendix A).

#### 3.1.2. Colostrum Intake

Colostrum intake was only measured at day 1. Therefore, age was not included as a fixed factor in the statistical analysis. None of the fixed factors (treatment, sex, and birth weight category) showed a significant interaction effect on colostrum intake. These interactions were removed as fixed factors from the linear mixed model, retaining only the individual fixed factors: treatment, sex, and birth weight. No effect of treatment (*p* = 0.575) (Figure 2) or sex (*p* = 0.295) (Appendix A) was observed for the colostrum intake. The VLBW piglets ingested significantly less colostrum than LBW piglets (*p* = 0.001) (Figure 2).

#### 3.1.3. Skin Lesion Scores

None of the fixed factors (treatment, age, sex, and birth weight category) showed a significant interaction regarding skin lesions scores. These interactions were removed as fixed factors from the linear mixed model, retaining only the individual fixed factors: treatment, age, sex, and birth weight. No treatment (*p* = 0.187) (Figure 3) or sex effect (*p* = 0.204) (Appendix A) was observed on the probability of having more severe skin lesions. Birth weight category had no effect on the severity of skin lesions (*p* = 0.295) (Figure 3). An age effect was observed (*p* < 0.0001). The highest risk of observing skin lesions was on day 24 (2 days after weaning), followed by day 1, day 2, day 9 and day 3 (Appendix A).

#### 3.1.4. Mortality

None of the fixed factors (treatment, age, sex, and birth weight category) showed a significant interaction in view of mortality. These interactions were removed as fixed factors from the linear mixed model, retaining only the individual fixed factors: treatment, age, sex, and birth weight. Treatment had no effect on the probability of dying (*p* = 0.572) (Figure 4). No sex effect was observed (*p* = 0.395) (Appendix A). The VLBW piglets had a significantly higher risk of dying than LBW piglets (Risk ratio: 2.39; *p* < 0.001) (Figure 4). The animals had the greatest risk of dying during the first day after birth, with the risk decreasing over the following time points at Farm A (*p* < 0.001) (Figure 4).

### 3.2. Experiment 2

#### 3.2.1. Body Weight and Growth

The performance data of LBW piglets receiving the 5 treatments at Farm B were similar to those observed at Farm A in experiment 1. There were no significant interaction effects between treatment, age, and sex on the variables related to body weight and growth. Therefore, only the individual fixed factors of treatment, age, and sex were retained. Treatment did not affect the body weight (*p* = 0.345) (Figure 5), ADG (*p* = 0.207), factorial growth (*p* = 0.241), metabolic weight (*p* = 0.307) or factorial metabolic rate (*p* = 0.477) (Appendix A). There was no difference between male and female LBW piglets for any of the parameters: body weight: *p* = 0.716, ADG: *p* = 0.301, factorial growth: *p* = 0.602, metabolic weight: *p* = 0.812, factorial metabolic rate *p* = 0.777 (Appendix A). Body weight increased over time (*p* < 0.001) (Figure 5), as did the factorial growth (*p* < 0.001), metabolic weight (*p* < 0.001) and factorial metabolic rate (*p* < 0.001) (Appendix A).

In comparing both farms, none of the fixed factors (treatment, age, sex) showed a significant interaction with ‘farm’. In the analysis of the individual fixed factors, only age (see above) had a significant impact. Thus, no difference in the LBW piglets’ body weight (*p* = 0.439) (Figure 1, 5), ADG (*p* = 0.062), factorial growth (*p* = 0.095), metabolic weight (*p* = 0.051) or factorial metabolic rate (*p* = 0.956) was observed when comparing Farm A with Farm B (Appendix A).

#### 3.2.2. Colostrum Intake

As in experiment 1, there were no significant interaction effects on colostrum intake between treatment, age, and sex. Therefore, only the individual fixed factors of treatment and sex were retained. In addition, at Farm B, colostrum intake did not differ significantly between the different treatment groups (*p* = 0.277) (Figure 6) or between males and females (*p* = 0.825) (Appendix A).

In comparing both farms, none of the fixed factors (treatment, age, sex) showed a significant interaction with ‘farm’. Next to treatment and sex, there was no difference in the LBW piglets’ colostrum intake between Farm A (*n* = 38) and Farm B (*n* = 33) (*p* = 0.421) (Figure 2, 6; Appendix A).

#### 3.2.3. Skin Lesion Scores

As in experiment 1, none of the fixed factors (treatment, age, and sex) showed a significant interaction effect on skin lesion scores. These interactions were removed as fixed factors from the model, retaining only the individual fixed factors: treatment, age, and sex, which showed similar effects as in experiment 1. Treatment did not affect the risk of having more severe skin lesions (*p* = 0.352) (Figure 7). No sex effect was observed either (*p* = 0.364) (Appendix A). An age effect was observed (*p* < 0.001): the highest risk of observing skin lesions was on day 26 (2 days after weaning), followed by day 9, day 3, day 2, and day 1 (Appendix A).

When comparing both farms, none of the other fixed factors (treatment, age, and sex) showed an interaction with farm as a fixed factor. However, at Farm A, the probability of having more severe skin lesions was higher compared to Farm B (*p* < 0.001) (Figure 3 and Figure 7).

#### 3.2.4. Mortality

As in the data on mortality in experiment 1, the data from experiment 2 showed no significant interactions between treatment, age, and sex. Thus, in the statistical analysis, only the fixed factors were retained. As in experiment 1, treatment did not affect the mortality of LBW piglets (*p* = 0.999) (Figure 8). No sex effect was observed either (*p* = 0.886) (Appendix A). There was an age effect that affected the risk of dying (*p* < 0.001). The highest risk of dying was between day 3 and day 9 (*p* < 0.001).

When comparing both farms (no interaction was observed for the other fixed factors), the risk of dying for LBW piglets was significantly higher at Farm A (Risk ratio 10.05; *p* < 0.001) (Figure 4 and Figure 8).

## 4. Discussion

This study was designed to evaluate whether the oral supplementation of a dense, concentrated milk replacer to LBW piglets affected their performance (growth, survival). Simultaneously, the existence of a lower limit (in terms of birth weight) up to which drenching would have an effect was assessed. The rationale was that, by concentrating the milk replacer, more energy and nutrients could be provided to the LBW piglets per supplementation and would provide the piglets with the necessary boost to start ingesting more colostrum and milk themselves.

The results from our first experiment showed no effect of drenching DMR on either the LBW or VLBW piglets’ growth or survival, and thus, a very low birth weight (<750 g) cannot be used as a criterion to refine the target group for drenching. However, the overall performance and survival of VLBW piglets, even after one or three doses of DMR, were much lower than that of LBW piglets. This suggests that VLBW piglets might be too weak to benefit from interventions such as drenching, and consequently, should not be considered for supplementation. This accords with Paredes et al. [39]. They concluded that piglets that deviate more than 2.5 SD from the mean birth weight of the farm population exhibit no potential to compensate their growth and performance under practical farm conditions. On average, VLBW piglets weighed 26% less than LBW piglets at birth and reduced the weight difference to 17% on the second day post-weaning. This was illustrated by the fact that LBW piglets increased their weight by a factor of 4.7 on average from birth to weaning, while VLBW piglets increased their body weight by a factor of 5.2. However, no statistically significant difference in growth was observed. Whereas some authors did not see any compensatory growth [40,41,42], others observed an increased body weight gain (relative to their birth weight) in lighter piglets [3], depending on their body mass index [13]. However, the aforementioned studies applied different definitions or selection protocols for (V)LBW piglets, based solely on the individual birth weight [3,40], the individual and litter birth weight [41], the morphology [13], or in some instances, even removed piglets with a birth weight below 750 g from the experiment [40,42], making it difficult to compare. Although the compensatory growth of VLBW piglets remains debatable, our results showed an explicitly inferior performance compared to LBW piglets. The VLBW piglets consumed considerably lower amounts of colostrum (below the required 250 g to survive [5]). They were characterized by very high mortality rates, mainly during the first week after birth. The cumulative mortality of VLBW piglets rose to 68%, whereas that of LBW piglets was limited to 30% at the farm with low perinatal farrowing management. Similar differences in mortality between these birth weight categories were found by Quiniou et al. [3]. In that study, the post-weaning mortality of piglets with a birth weight between 610 g and 800 g rose to 52%, whereas that of piglets, born between 810 g and 1 kg, was limited to 29%. Contrary to our study, Quiniou and colleagues [3] found lower mortality rates in the very small piglets, which could be attributed to the applied management strategies, such as cross-fostering and the provision of a heat lamp.

When considering the LBW piglets (piglets with a birth weight between 750 g and 1 kg and not deviating more than 1.5 SD from the litter’s mean birth weight), the DMR did not affect any of the assessed parameters, regardless of whether the animals were drenched once or three times. Following these results, another study in which DMR with a similar energy density was supplemented to piglets did not find an improved colostrum intake or survival. However, the authors did observe an increased body weight and small intestine development in the DMR-fed piglets at weaning age (21 days) [43]. In contrast to the current study (60 kJ per dose of 5 mL DMR), the authors provided the DMR ad libitum to the piglets twice a day, enabling the piglets to acquire a higher caloric intake. The importance of providing enough calories during the first hours after birth was illustrated in similar studies where fat-based supplements were drenched, and no effect on mortality was found when only 71–74 kJ was given to the animals [17]. In another study by Declerck and colleagues [18], a decrease in mortality was observed in piglets below 1 kg when the piglets were provided with 156–165 kJ within the first day after birth. Furthermore, the piglets used in the study by de Greeff et al. [43] had an average birth weight of 1.3 kg, whereas the LBW piglets in our experiments had an average birth weight of 0.86 kg. Thus, not only the amount of ingested DMR but also the difference in birth weight can explain these contradictory results, since small, low-viability piglets often have impaired gut development and nutrient absorption [44]. Consequently, LBW piglets require not only a high-caloric energy boost and nutrient supplementation, but also a supplement that can improve gut function (reviewed by [44]), a factor that is absent in a plain (concentrated) milk replacer.

A second experiment was conducted to evaluate the reproducibility of the results from experiment 1 at a farm with more intensive farrowing management. The comparison between two different management strategies would allow us to determine whether a lack of good perinatal care might act as a confounding factor and might mask any potential effect of drenching DMR. It was hypothesized that in the case of a higher level of perinatal care (supervision, supplemental heat), DMR improved the performance of LBW piglets more than in the case of a lower level of perinatal care.

When comparing the results of LBW piglets between the two farms, the birth weight and subsequent growth performance did not differ, regardless of whether the animals were drenched with DMR once or three times or not supplemented at all. Thus, it would appear the farm’s perinatal management did not influence any potential effect of the DMR. However, the mortality and the risk of skin lesions of LBW piglets were significantly lower at the farm with more labour-intensive perinatal management. Additionally, despite no effect of DMR on mortality at any of the farms, there was more variation between treatment groups at the farm with low perinatal care. Good farrowing and neonatal management have been shown to have an important effect on the survival of piglets. Nevertheless, little is known about good perinatal care for (very) small piglets since most studies and reviews have focused on piglets with a birth weight above 1 kg [23,44,45,46,47,48,49,50]. Therefore, it remains difficult to attribute the higher survival and lesser skin lesions at the second farm to one distinct management strategy. Moreover, the differences in genetics, feeding strategy, etc. could have influenced these results as well. An important difference between the two farms was the presence of staff during farrowing at Farm B. This was possible since most sows farrowed in one day, via the use of farrowing induction. At Farm A, farrowing was much less induced, and sows farrowed during a period of three days. Farrowing supervision, accompanied by regular drying and assistance to the udder, can improve the colostrum intake and the survival of piglets, increasing the survival chances of LBW piglets [23,44,46,47,51]. However, no difference in colostrum intake was observed between the two farms. The mechanistic model by Theil et al. [31] requires the time of birth of the litter’s first piglet, the birth weight of the piglet of interest, and the body weight of the latter, 24 h after the birth of the first piglet. Potentially, the low number of piglets we could include in the calculation can explain why we could not detect a difference in colostrum intake between the two farms. A higher colostrum intake was to be expected at the farm with higher perinatal care, given the occasional drying and assistance to the teat. Secondly, no cross-fostering of the LBW piglets was performed at Farm A. Whereas some authors found a positive effect of cross-fostering on growth [45] or survival [50], others found no positive results [50,52]. This, combined with a lack of knowledge on the exact effect of cross-fostering on LBW piglets, often results in the application of intermediate protocols in practice [46]. Another management strategy that was only applied at Farm B was the supplementary feeding of a milk replacer via milk dishes (starting 48 h after farrowing). The positive effect of providing a milk replacer is also debatable. Some studies have shown a positive effect on growth and survival [45], while others did not observe such an effect [49]. In general, it remains difficult to attribute an exact effect of one specific intervention on the performance and survival of piglets. More research, mainly on LBW piglets, is needed regarding the effect of individual or combined perinatal management strategies.

## 5. Conclusions

The present study found a significantly lower performance and survival of (V)LBW piglets at the farm with low perinatal care. Drenching DMR had no effect on the performance or survival of LBW piglets, regardless of the quality of the perinatal care. It is challenging to provide LBW piglets with enough calories within a practically-achievable number of drenching applications. Good farrowing management and neonatal care improved the survival level, but not growth performance, of LBW piglets. It remains difficult to attribute this positive effect to one or more interventions. Our experimental set-up only allowed us to evaluate the perinatal management as a confounding factor on the effect of drenching. Thus, more research on a good perinatal protocol for LBW piglets is needed, as it appears to be more beneficial for LBW piglets than supplementing a dense milk replacer.

## Figures and Tables

**Figure 1 animals-13-00063-f001:**
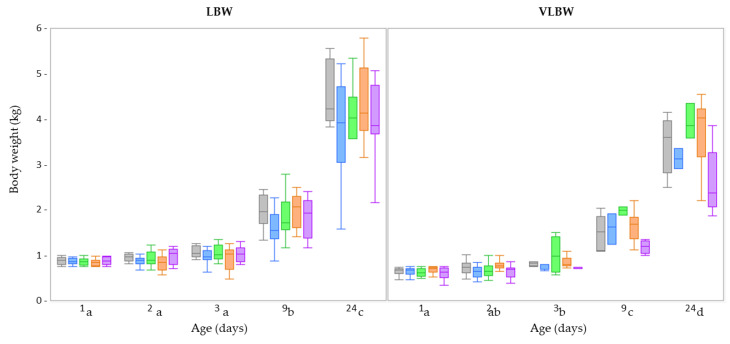
Boxplots of the body weight at different time points (day of birth [day 1], day 2, day 3, day 9, and day 24 [two days after weaning]) of very low birth weight (VLBW; *n* = 80) and low birth weight (LBW; *n* = 80) piglets per treatment (dense milk replacer [DMR] one dose [grey box], DMR three doses [blue box], no treatment [green box], sham one dose [orange box], sham three doses [purple box]) at Farm A. There was no effect of drenching DMR on body weight (*p* = 0.179). Body weight increased during the experimental period independent of treatment and birth weight category (*p* < 0.001). Ages carrying a different subscript letter were significantly different.

**Figure 2 animals-13-00063-f002:**
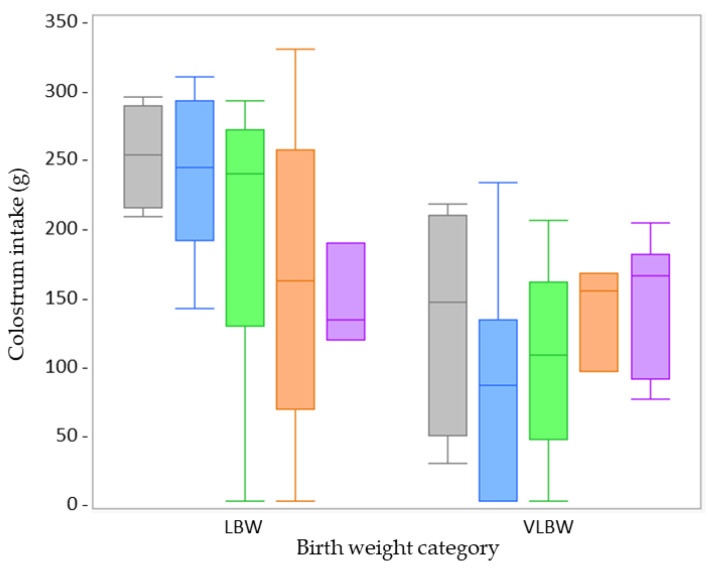
Boxplots of the colostrum intake at Farm A from low birth weight (LBW; *n* = 38) and very low birth weight (VLBW; *n* = 33) piglets per treatment (dense milk replacer [DMR] one dose [grey box; LBW *n* = 4, VLBW *n* = 5], DMR three doses [blue box; LBW *n* = 11, VLBW *n* = 9], no treatment [green box; LBW *n* = 9, VLBW *n* = 6], sham one dose [orange box; LBW *n* = 11, VLBW *n* = 3], sham three doses [purple box; LBW *n* = 3, VLBW *n* = 10]). There was no effect of treatment on colostrum intake (*p* = 0.575). Irrespective of treatment, colostrum intake was significantly less in VLBW when compared to LBW piglets (*p* = 0.001).

**Figure 3 animals-13-00063-f003:**
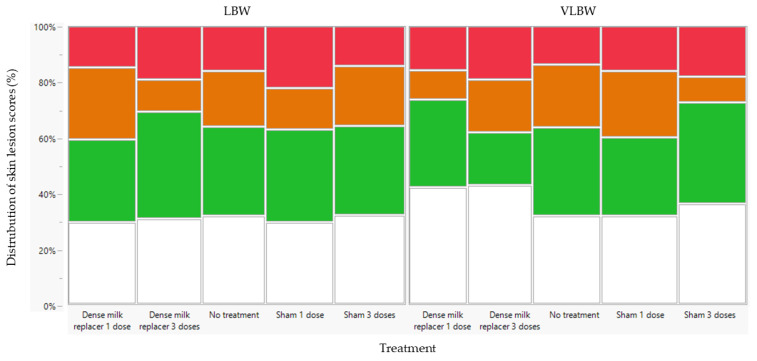
Distribution of skin lesion (SL) scores at Farm A (low perinatal management) of the selected low birth weight (LBW, *n* = 80) and very low birth weight (VLBW, *n* = 80) piglets per treatment (dense milk replacer [DMR] one dose, DMR three doses, no treatment, sham one dose, sham three doses). There was a no significant effect of treatment or birth weight category on the probability of having more severe SL. The following scoring system was applied: 0: no lesions (white); 1: <5 superficial lesions (skin unbroken) (green); 2: 5–10 superficial lesions or <5 deep lesions (skin broken and evidence of haemorrhage) (orange); 3: >10 superficial lesions or >5 deep lesions (red).

**Figure 4 animals-13-00063-f004:**
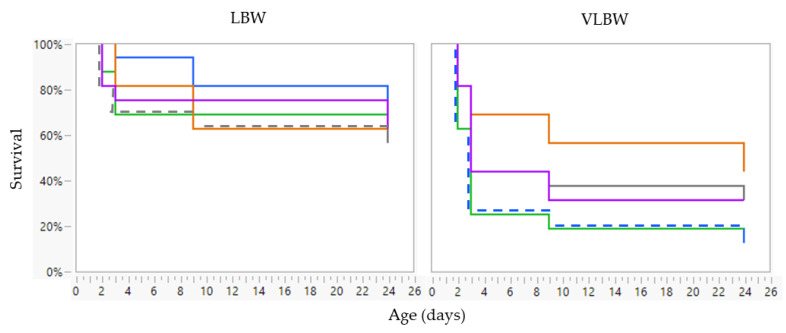
Cumulative mortality at Farm A of very low birth weight (VLBW; *n* = 80) and low birth weight (LBW; *n* = 80) piglets per treatment (no treatment (green line), sham 1 dose (orange line), sham 3 doses (purple line), dense milk replacer 1 dose (grey line) or dense milk replacer 3 doses (blue line)). Cox’s proportional hazard model showed that VLBW piglets had a significantly higher risk of dying than LBW piglets (Risk ratio: 2.39; *p* < 0.001). The animals had the greatest risk of dying during the first day after birth, with the risk decreasing over the following time points (*p* < 0.001).

**Figure 5 animals-13-00063-f005:**
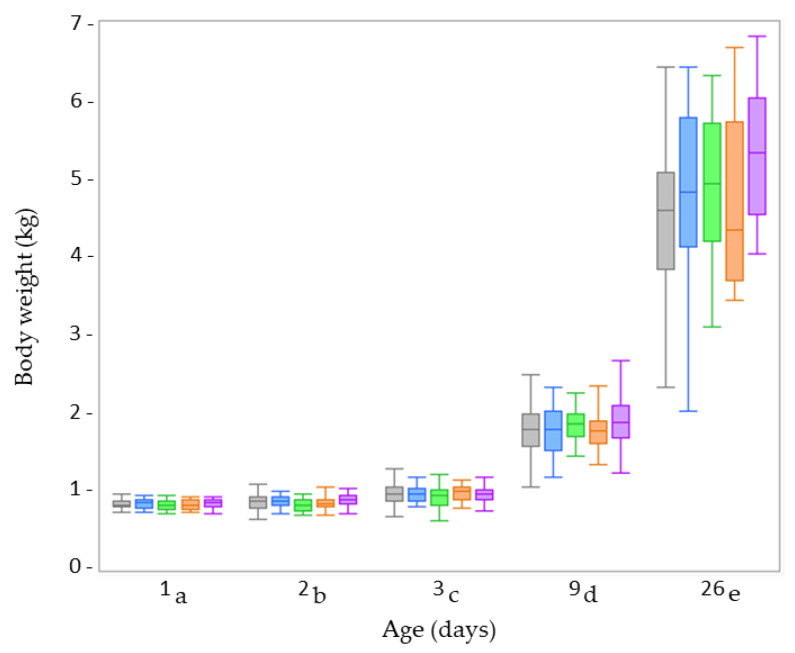
Boxplots of the body weight at different time points (day of birth [day 1], day 2, day 3, day 9, and day 26 [two days after weaning]) of low birth weight (LBW; *n* = 150) piglets per treatment (dense milk replacer [DMR] one dose [grey box], DMR three doses [blue box], no treatment [green box], sham one dose [orange box], sham three doses [purple box]) at Farm B. There was no effect of drenching DMR on body weight (*p* = 0.345). Body weight increased during the experimental period independent of treatment and birth weight category (*p* < 0.001). Ages carrying a different subscript letter were significantly different.

**Figure 6 animals-13-00063-f006:**
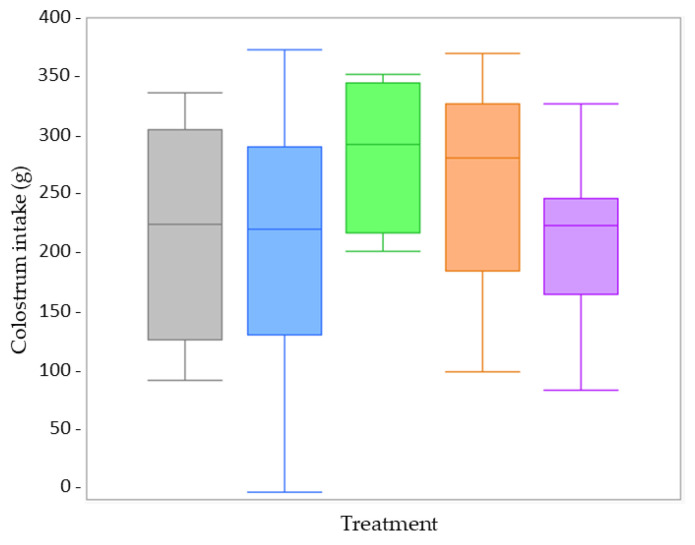
Boxplots of the low birth weight piglets’ colostrum intake per treatment (dense milk replacer [DMR] one dose [grey box], DMR three doses [blue box], no treatment [green box], sham one dose [orange box], sham three doses [purple box]) at Farm B (high perinatal management; *n* = 33). Colostrum intake did not differ significantly between the different treatment groups (*p* = 0.277).

**Figure 7 animals-13-00063-f007:**
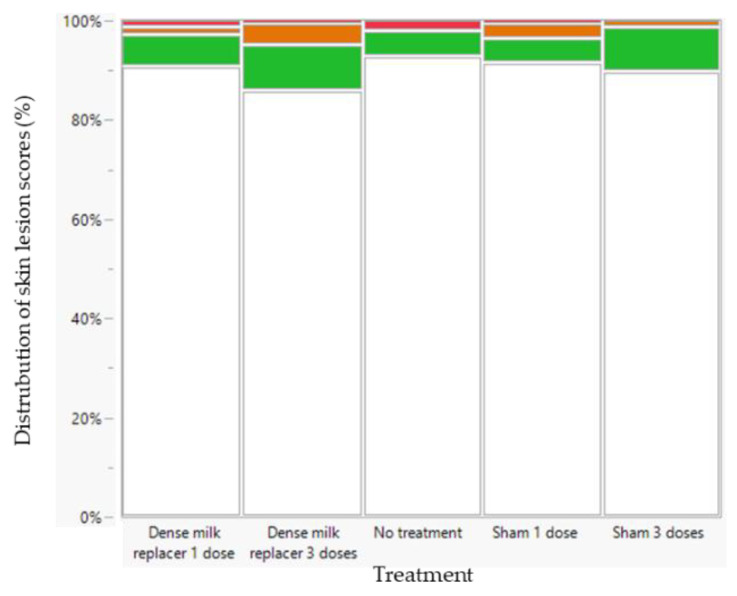
Distribution of skin lesion (SL) scores per treatment at Farm B (low perinatal management) of the selected low birth weight piglets (*n* = 150) per treatment (dense milk replacer [DMR] one dose, DMR three doses, no treatment, sham one dose, sham three doses). The following scoring system was applied: 0: no lesions (white); 1: <5 superficial lesions (skin unbroken) (green); 2: 5–10 superficial lesions or <5 deep lesions (skin broken and evidence of haemorrhage) (orange); 3: >10 superficial lesions or >5 deep lesions (red).

**Figure 8 animals-13-00063-f008:**
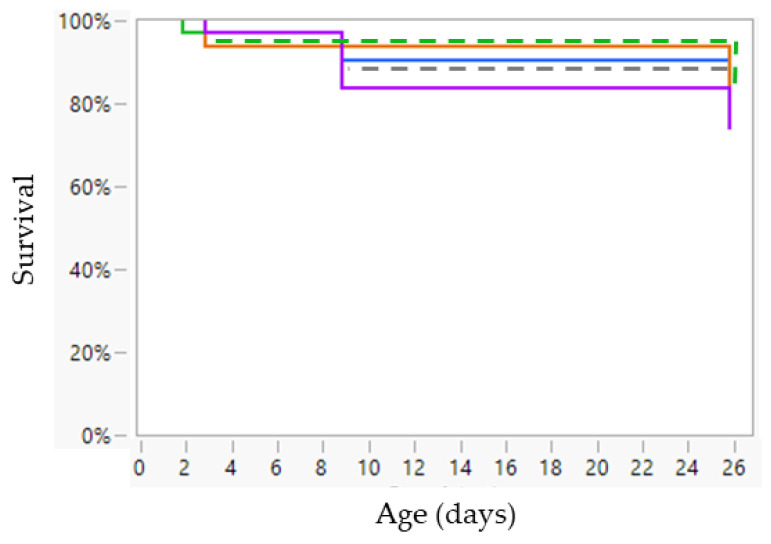
Cumulative mortality of low birth weight (LBW) piglets from Farm B (n = 150) at different time points (day 1, 2, 3, 9, and two days after weaning [day 26]) per treatment: no treatment (green line), sham one dose (orange line), sham three doses (purple line), dense milk replacer one dose (grey line) or dense milk replacer three doses (blue line). Cox’s proportional hazard model showed no effect of treatment (*p* = 0.999). The animals had the greatest risk of dying between day 3 and 9 (*p* < 0.001).

**Table 1 animals-13-00063-t001:** Farrowing management at the two farms, the sows, the very low birth weight (VLBW), and low birth weight (LBW) piglets that were used during the study.

Parameter	Farm A	Farm B
Breed (sow)	TN70: Topigs Norsvin x Norsvin Landrace	Danbred YL hybrid: Danbred Yorkshire x Danbred Landrace
Parity (mean ± SD)	4.31 ± 2.52	3.23 ± 1.67
Average litter size (mean ± SD)	17.14 ± 2.96	19.92 ± 3.24
Birth weight VLBW piglets (kg; mean ± SD)	0.64 ± 0.10	Not applicable
Birth weight LBW piglets (kg; mean ± SD)	0.86 ± 0.07	0.87 ± 0.06
Birth weight all weighed piglets (BW_population_; kg; mean ± SD) *	1.28 ± 0.35	1.17 ± 0.32
Weaning age (days; mean ± SD) **	20 ± 0.79	23 ± 0.00
Farrowing induction	Only in case of prolonged or no labour	All sows
Monitoring during farrowing	Twice a day (morning and evening round)	Constant supervision throughout day
Drying new-born piglets	No	Sometimes
Assistance first suckle	No	Sometimes
Heat provision	No, plastic cover over creep area	Heated floor in creep area
Cross-fostering	No	Yes
Milk supplementation	No	Yes
Hygiene lock	Yes, no shower	Yes, with shower
Selected piglets	160	150

* BW_population_ = all piglets (normal, LBW, and VLBW) that were born from 139 and 230 sows at farms A and B, respectively; ** During every farrowing round, sows farrowed over a time course of 3 days at Farm A, while all piglets were born and selected on the same day at Farm B (all sows farrowed in the course of 1 day).

**Table 2 animals-13-00063-t002:** Nutrient, chemical, and energetic composition of the supplemented milk replacer.

Analytical Constituents		Nutritional Additives		
Crude protein (%)	19.9	Vitamin A (IU/kg)		25,000
Crude fat (%)	15.9	Vitamin D3 (IU/kg)		5000
Crude ash (%)	7.6	Vitamin E (mg/kg)		80
Crude fibre (%)	0	Vitamin K (mg/kg)		4
Moisture (%)	3.1	Vitamin C (mg/kg)		158
Lactose (%)	38.5	Vitamin B1 (mg/kg)		6
Lysine (%)	1.75	Vitamin B2 (mg/kg)		6
Methionine (%)	0.62	Vitamin B6 (mg/kg)		4
Cystine + Methionine (%)	1	Vitamin B12 (µg/kg)		40
Calcium (%)	0.55	Iodine (mg/kg)		1
Sodium (%)	0.62	Manganese (mg/kg)		45
Phosphorus (%)	0.5	Zinc (mg/kg)		84
Magnesium (%)	0.12	Selenium (mg/kg)		0.30
Iron (mg/kg)	76	Propyl gallate (mg/kg)		3
Copper (mg/kg)	155	Butylated hydroxyanisole (mg/kg)		3
Energetic value				
Metabolizable energy (MJ/kg|kcal/kg)			17.9|4280	
Net energy (MJ/kg|kcal/kg)			14.3|3420	

**Table 3 animals-13-00063-t003:** Comparison of average daily growth (ADG), factorial growth, metabolic weight and factorial metabolic rate of low birth weight (LBW) with very low birth weight (VLBW) piglets at days 1, 2, 3, 9 and 2 days post-weaning (24 days) at the farm with low perinatal management (Farm A (median ± SD)).

		VLBW	*n*	LBW	*n*	*p*-Value
**ADG (kg)**	Day 2	0.10 ± 0.15	26	0.08 ± 0.12	56	0.783
	Day 3	0.10 ± 0.06	26	0.10 ± 0.07	56	0.854
	Day 9	0.11 ± 0.05	26	0.12 ± 0.05	55	0.274
	Day 24	0.12 ± 0.03	24	0.14 ± 0.04	53	**0.022**
**Factorial growth**	Day 2	1.02 ± 0.22	59	1.06 ± 0.18	72	0.303
	Day 3	1.14 ± 0.31	33	1.20 ± 0.22	62	0.938
	Day 9	2.25 ± 0.63	26	2.12 ± 0.56	56	0.828
	Day 24	4.91 ± 1.31	24	4.94 ± 1.39	53	0.957
**Metabolic weight (kg^0.75^)**	Day 1	0.74 ± 0.08	80	0.89 ± 0.10	80	**<0.001**
	Day 2	0.77 ± 0.12	59	0.93 ± 0.13	72	**<0.001**
	Day 3	0.85 ± 0.13	33	1.01 ± 0.15	62	**0.002**
	Day 9	1.44 ± 0.25	26	1.56 ± 0.27	56	**0.001**
	Day 24	2.54 ± 0.43	24	2.84 ± 0.52	53	**0.018**
**Factorial metabolic rate**	Day 2	1.01 ± 0.16	59	1.05 ± 0.13	72	0.303
	Day 3	1.11 ± 0.21	33	1.15 ± 0.16	62	0.938
	Day 9	1.84 ± 0.38	26	1.76 ± 0.38	56	0.828
	Day 24	3.30 ± 0.66	24	3.31 ± 0.82	53	0.957

## Data Availability

All data presented in this study is contained within the article.

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
