# Peer review of "The Effect of Drenching (Very) Low Birth Weight Piglets with a Dense, Concentrated Milk Replacer at Farms with Differing Farrowing Management"

_animals, 2022, doi:10.3390/ani13010063_

Round 1
Reviewer 1 Report
Simple summary:
Line 18-20: Would suggest re-wording, the current wording could be interpreted as implying the expectation of an effect.
Introduction:
Line 72-76: Similar comment as for the simple summary.
Materials and methods:
Table 1: Would be good to include average litter size (at farrowing and after any cross-fostering). Farrowing duration is sometimes also used as referring to the time from birth of the first to last piglet within a litter, so including a definition for the use here would be helpful.
Line 156-157: Are days 24 and 26 the days of weaning or the 2 days after weaning?
Line 216-217, 250-251: Piglet is the experimental unit and shouldn't be in the statistical model. If it is included, it should be removed, or if I am interpreting this incorrectly please re-word to clarify.
Statistical analysis, general: It would be good to include the final model for each variable, perhaps as an appendix table.
Results:
Figures: The legend is in the text, but would be better included visually to help the figures be stand-alone.
Line 282: References figure 1A, but figure is not labeled with A or B
General: For results that are referenced but not presented (e.g., line 280 for DMR effects), it would help to say data not presented, as other results reference tables and figures, and it could be confusing if the reader expects these results to be in the previously mentioned table or figure as well.
Check table and figure footnotes for italicizing p for p-values.
Line 343: Figure 4 refers only to piglet body weight, is this statement of no interaction between farm and treatment only applicable to piglet body weight, or all measurements?
Line 355: Same comment as for line 343, only relating to colostrum intake, or all measurements?
Line 386-387: Would re-word to state that there were no interactions between farm and treatments for piglet mortality, as there could be more non-measurable differences between farms than just management (e.g., weather, disease status, etc.).
Figure 8: Alignment is different from the other figures.
Discussion:
Line 448-450: While this may be a good point to debate, the LBW piglets also did not show a beneficial effect of drenching. Is this point of being too weak to benefit also applicable to these LBW piglets, or are other there other potential mechanisms such as reduced colostrum intake due to satiation?
Author Response
Response to reviewers
Manuscript ID: animals-202948483
Manuscript title: The effect of drenching (very) low birth weight piglets with a dense, concentrated milk replacer at farms with differing farrowing management
The authors would first of all like to thank the academic editor and the reviewers for the constructive comments. We have made every effort to include and address all concerns that were pointed out. We hope the improvements to the manuscript are to your liking, but if further clarifications or actions on our part are required, we are at your disposal.
Reviewer 1
L-18-20: Would suggest re-wording, the current wording could be interpreted as implying the expectation of an effect.
We agree with the reviewer and have rephrased the sentence.
L-18-20: This study aimed to determine if the performance of low birth weight piglets can be improved by drenching a dense, concentrated milk replacer and whether the frequency of drenching and the severity of the low birth weight played a role.
L-72-76: Similar comment as for the simple summary.
The sentences have been rephrased.
L-75-81: The first experiment examined whether drenching a dense milk replacer affected the performance (growth, survival) of low birth weight piglets and what frequency of drenching was optimal. It was hypothesized that the milk replacer would improve their performance indirectly by supplying enough energy so the piglet can achieve a (first) suckle. Additionally, it was tested whether this boosting effect would be higher when the milk replacer was drenched three times versus only once. Simultaneously the existence of a lower limit – in terms of birth weight – up to which drenching would have an effect, was assessed.
Table 1: Would be good to include average litter size (at farrowing and after any cross-fostering). Farrowing duration is sometimes also used as referring to the time from birth of the first to last piglet within a litter, so including a definition for the use here would be helpful.
Thank you for this remark. Indeed, how it was mentioned (farrowing duration) in the table was strange. We have removed this parameter from the table but have included an explanation (since it determines the weaning age) in the explanation of the weaning age. It also relates to the use of farrowing induction. This is added to the table.
L-136-137**During every farrowing round, sows farrowed over a time course of 3 days at farm A, while all piglets were born and selected on the same day at farm B (all sows farrowed in the course of 1 day)..
We have also added the average litter size at farrowing in the table. Litter size after cross-fostering was not registered (at farm B, no cross-fostering was applied at farm A).
L-156-157: Are days 24 and 26 the days of weaning or the 2 days after weaning?
These are the 2 days after weaning. We have rephrased this.
L-167: All piglets were weighed on the day of birth (day 1), day 2, day 3, day 9, and two days after weaning (day 24 and day 26 at farms A and B, respectively).
L-216-217, 250-251: Piglet is the experimental unit and shouldn't be in the statistical model. If it is included, it should be removed, or if I am interpreting this incorrectly, please re-word to clarify.
Indeed, piglet is the experimental unit. We have opted to randomize the treatments across the piglets, not considering the sow in this respect, but including sow as a random factor. Repeated measurements were performed on the piglets. Therefore, piglets, nested in sow were included in the statistical model. We have clarified this in the text.
L-222-225: To account for the dependence between littermates or between measurements on the same piglet (but at a different age) random factors for sow (nested in the farrowing round) and piglet (nested in sow) were included.
Statistical analysis, general: It would be good to include the final model for each variable, perhaps as an appendix table.
We shortened the description of the statistical analysis to increase its clarity and have recapitulated the different decisions of the stepwise backward modelling in the results. Additionally, we have separated the statistical analysis for both experiments to clarify the applied approach to the reader.
Figures: The legend is in the text but would be better included visually to help the figures be stand-alone.
We have added more information in the legends to make the figures and tables more stand alone.
Line 282: References figure 1A, but figure is not labeled with A or B
This was a mistake. We have deleted the A.
General: For results that are referenced but not presented (e.g., line 280 for DMR effects), it would help to say data not presented, as other results reference tables and figures, and it could be confusing if the reader expects these results to be in the previously mentioned table or figure as well.
We have added a table as supplementary material including all the medians ± SD’s for all the subgroups within the experiments.
Check table and figure footnotes for italicizing p for p-values.
We have performed the needed corrections.
L-343: Figure 4 refers only to piglet body weight, is this statement of no interaction between farm and treatment only applicable to piglet body weight, or all measurements?
We have rephrased the results.
L-368-371: The performance data of LBW piglets receiving the 5 treatments at farm B were similar as those observed at farm A in experiment 1. There were no significant interaction effects between treatment, age, and sex on the variables related to body weight and growth. Therefore, only the individual fixed factors treatment, age, and sex were retained.
Figure 4 refers only to piglet body weight.
L-355: Same comment as for line 343, only relating to colostrum intake, or all measurements?
Here it only refers to colostrum intake. We have rephrased this;
L-396-398: As in experiment 1, there were no significant interaction effects on colostrum intake between treatment, age, and sex. Therefore, only the individual fixed factors of treatment and sex were retained.
L-386-387: Would re-word to state that there were no interactions between farm and treatments for piglet mortality, as there could be more non-measurable differences between farms than just management (e.g., weather, disease status, etc.).
We have reworded this in view of other remarks and questions on the statistical analysis. We hope this clarifies it. In the discussion, we comment on the fact that the differences observed between both the farms could be more than solely perinatal management.
L-434-442: As in the data on mortality in experiment 1, the data of experiment 2, showed no significant interactions between treatment, age, and sex. Thus, in the statistical analysis only the fixed factors were retained. As in experiment 1, treatment did not affect the mortality of LBW piglets (p = 0.999) (Figure 8). No sex effect was observed either (p = 0.886) (Supplementary material, Figure S7). There was an age effect that affected the risk of dying (p < 0.001). The highest risk of dying was between day 3 and day 9 (p < 0.001).
When comparing both farms (no interaction was observed for the other fixed fac-tors), the risk of dying for LBW piglets was significantly higher at farm A (Risk ratio 10.05; p < 0.001) (Figure 4, 8).
Figure 8: Alignment is different from the other figures.
We have update and realigned the figures.
L-448-450: While this may be a good point to debate, the LBW piglets also did not show a beneficial effect of drenching. Is this point of being too weak to benefit also applicable to these LBW piglets, or are other there other potential mechanisms such as reduced colostrum intake due to satiation?
Indeed, LBW piglets did not benefit either from the DMR supplementation in the used dosages. However, in the referenced study, where the authors used higher dosages, positive effects were observed in LBW piglets.
Piglets that belong to the VLBW category have much higher mortality rates and lack the ability to compensate for their lower performance, in contrast with LBW piglets. This difference in potential compensation can possibly explain why LBW piglets are not too weak to benefit from supplementation, whereas VLBW do not have this potential.
A satiation effect was not expected here, given the colostrum intake results and low dosages/volumes.
Reviewer 2 Report
General Comments.
Two experiments (field trials) were conducted to evaluate effects of dosing low birth weight (LBW) and very low birth weight (VLBW) pigs with a concentrated milk replacer. After no differences due to dosing protocols were detected on the first farm, which was classified as a low perinatal management farm, a second farm was included which was classified as a farm with high quality perinatal care. Differences in survival of LBW and VLBW were detected, however beneficial effects from the milk replacer were not detected. The authors discussed several reasons that beneficial effects were not detected. One explanation that may need additional consideration is the amount of energy supplied by the milk replacer dose. See specific comments below on the calculations of energy supplied.
Specific comments:
L 98. Was parturition natural or hormone-induced? (this is mentioned in L 496, but should be defined in the methods).
L 140. I need some help with the energy calculations. If I assume 6 g milk replacer X 0.969 g DM/g = 5.814 g DM diluted into 4 ml H20 = 1.4535 g DM/ mL X 5 mL dose = 7.2675 g DM or 0.007265 kg X 17.9 MJ/kg *0.969 =0.126 MJ per dose = 126 kJ ME/dose. This calculation does not consider the change in density of the diluted 6g/4 mL in the original dose.
Your calculations state that 59.61 kJ ME was provided. If your calculations are correct and the pig requires ~ 200 kcal (From Odle et al., 1989. J. Anim. Sci 67:3340-3351) or ~ 800 kJ, then the energy supplied less than 10% of the pigs daily energy needs. Quite a low dose to induce a response. High dosages of a concentrated carbohydrate source may create osmotic loads in the GI tract. Based on the survival data, no negative responses (nor positive) were detected.
L 135-144. What was the temperature of the 5 mL dose at the time of dosing? Apparently the dose was mixed at 40 C, was each dose mixed at the time of dosing the individual pig or were larger batches mixed and perhaps refrigerated until dosing?
L 172. The calculation of colostrum intake is simply a function of gain and body weight. Was this equation derived from LBW and VLBW pigs? As body composition is not a component of the model the application to the current efforts are questionable.
L 190 to 205. With acknowledgement that saliva samples could not the collected at the early ages, when the milk replacer was likely most likely to be beneficial, collections of samples at 47 days of age is unlikely a reliable predictor of the same stress response in a 1-day old pig. These measures should be omitted.
Figure 10. The legend should acknowledge that samples were collected from pigs at 47 days of age. The inferences from these responses relative to dosages over the first week of life are questionable.
L 522. “It is challenging to provide LBW piglets with enough calories within a practically achievable number of drenching applications” If the energy calculations above are correct, I agree. Other approaches, such as medium chain triglycerides, are needed to get the energy dosages to sufficient levels. Jack Odle has contributed a substantial amount of work in this area over his career. (may be of benefit for your future efforts)
Author Response
Response to reviewers
Manuscript ID: animals-202948483
Manuscript title: The effect of drenching (very) low birth weight piglets with a dense, concentrated milk replacer at farms with differing farrowing management
The authors would first of all like to thank the academic editor and the reviewers for the constructive comments. We have made every effort to include and address all concerns that were pointed out. We hope the improvements to the manuscript are to your liking, but if further clarifications or actions on our part are required, we are at your disposal.
Point-by-point review:
Reviewer 2
L-98: Was parturition natural or hormone-induced? (this is mentioned in L 496, but should be defined in the methods).
We have added this to the table with more information on both farms.
L-140: I need some help with the energy calculations. If I assume 6 g milk replacer X 0.969 g DM/g = 5.814 g DM diluted into 4 ml H20 = 1.4535 g DM/ mL X 5 mL dose = 7.2675 g DM or 0.007265 kg X 17.9 MJ/kg *0.969 =0.126 MJ per dose = 126 kJ ME/dose. This calculation does not consider the change in density of the diluted 6g/4 mL in the original dose.
Your calculations state that 59.61 kJ ME was provided. If your calculations are correct and the pig requires ~ 200 kcal (From Odle et al., 1989. J. Anim. Sci 67:3340-3351) or ~ 800 kJ, then the energy supplied less than 10% of the pigs daily energy needs. Quite a low dose to induce a response. High dosages of a concentrated carbohydrate source may create osmotic loads in the GI tract. Based on the survival data, no negative responses (nor positive) were detected.
The energetic value of a 5 mL dose was estimated using the volume after dissolving the milk replacer in water. The end volume of 6 g milk replacer in 4 mL water, was exactly 9 mL. Thus, 5 mL of this solution contained 3.33 g of dry matter, equivalent to 17.9 kJ ME/g*3.33 g = 59.61 kJ ME.
This is indeed only a small fraction of the required 800 kJ, but it was one the study’s goals to test whether a small energetic booster (60 kJ when one dose was given or 180 kJ when three doses (within 24 h) were given) would suffice to provide small piglets with the required boost to achieve a first suckle. A small volume and fewer drenching applications would be more achievable for farmers.
We have tried to clarify the reasoning behind the low energetic booster in the introduction.
L-67-72: Moreover, in the case that drenching would significantly improve piglet performance, additional factors should be taken into account before drenching can be advised as a pre-weaning strategy, i.e. the labour costs and intensity (individual and/or repeated handling of supplemented piglets), and financial costs (supplemented products, spillage) associated with the nutritional support via drenching.
L-75-79: The first experiment examined whether drenching a dense milk replacer affected the performance (growth, survival) of low birth weight piglets and what frequency of drenching was optimal. It was hypothesized that the milk replacer would improve their performance indirectly by supplying enough energy so the piglet can achieve a (first) suckle. Additionally, it was tested whether this boosting effect would be higher when the milk replacer was drenched three times versus only once.
L-135-144. What was the temperature of the 5 mL dose at the time of dosing? Apparently the dose was mixed at 40 C, was each dose mixed at the time of dosing the individual pig or were larger batches mixed and perhaps refrigerated until dosing?
The DMR solution was made at the time of drenching. We have altered this in the text.
L-143-145: The DMR was prepared by dissolving 6 g of a plain milk replacer (Piglet Milk R714®, Table 2) in 4 mL of water (40°C) at the time of drenching.
L-172. The calculation of colostrum intake is simply a function of gain and body weight. Was this equation derived from LBW and VLBW pigs? As body composition is not a component of the model the application to the current efforts are questionable.
We have used the equation from Theil, which is frequently used to estimate colostrum intake based on body weight gain and duration of suckling. We do agree that the different body composition affecting body weight gain could obscure these measurements. However, given that we measure body weight gain over a period of 24 h immediately after birth, we feel these effects are negligible. DOI: 10.2527/jas.2014-7841
L-190 to 205. With acknowledgement that saliva samples could not the collected at the early ages, when the milk replacer was likely most likely to be beneficial, collections of samples at 47 days of age is unlikely a reliable predictor of the same stress response in a 1-day old pig. These measures should be omitted.
We have moved these measurements to the supplementary material.
Figure 10. The legend should acknowledge that samples were collected from pigs at 47 days of age. The inferences from these responses relative to dosages over the first week of life are questionable.
We have moved these measurements to the supplementary material.
L-522. “It is challenging to provide LBW piglets with enough calories within a practically achievable number of drenching applications” If the energy calculations above are correct, I agree. Other approaches, such as medium chain triglycerides, are needed to get the energy dosages to sufficient levels. Jack Odle has contributed a substantial amount of work in this area over his career. (may be of benefit for your future efforts)
Thank you sincerely for this useful tip!
Reviewer 3 Report
The paper dealt with a very interesting topic. The material and methods, experimental design are carefully thought and aspects such as the impact of drenching, the procedure itself, on animal welfare taken into account. The discussion is interesting and highlights the limitations of conducting experiments like this one in which it is difficult to standardise other factors that can have a clear influence on the results (other than the treatment being studied).
I only have minor comments to the manuscript, since according to my knowledge and opinion the authors have clearly explained the limitations encountered and how this could explain their results.
Table 1. What does it mean farrowing duration 3 days? Understood when reading the discussion, that it refers to the overall farrowing of all your sows. But it was a little bit confusing when reading only the Table.
Lines 127-130. Not totally clear to the reviewer. To account for a “sow” effect, there was one piglet receiving each treatment in each of the sows (ie 5 piglets controlled/sow, 2 drenching doses and 3 the sham treatments). Was this feasible, understanding that you also had to select LBW and VLBW piglets accross different sows? How was it balanced? Sorry I do not totally follow your explanation in Material and Methods. I see that this is somehow accounted for the statistical analysis, but it is not totally clear to the reviewer.
Line 206. One aspect that remains unclear is whether in the statistical analysis you accounted for the fact that you have repeated measures of parameters, for example for cortisol and chromogranin A. You mention that time was added as random effect. But actually what you have is a repeated measurement. How did you account for that?
Figure 1. Could you indicate somehow the statistical difference that you mention in the text in the Figure too? In line 282-283 you mention that in Figure 1A there is statistical difference at p<0.001 between LBW and VLBW, but then under the Figure you indicate “mixed models p<0.05”. It is a bit confusing. Same for Figure 2 and 4, lines in text and Figure do not indicate the same statistical difference. In Figure 5, text mentions no significant effect, whereas under the Figure again it appears “linear mixed models, p<0.05”
Figure 8. Maybe because of the pdf, the Chromogranin A graph is a bit cut in the manuscript I downloaded.
Author Response
Response to reviewers
Manuscript ID: animals-202948483
Manuscript title: The effect of drenching (very) low birth weight piglets with a dense, concentrated milk replacer at farms with differing farrowing management
The authors would first of all like to thank the academic editor and the reviewers for the constructive comments. We have made every effort to include and address all concerns that were pointed out. We hope the improvements to the manuscript are to your liking, but if further clarifications or actions on our part are required, we are at your disposal.
Point-by-point review:
Reviewer 3
Table 1. What does it mean farrowing duration 3 days? Understood when reading the discussion, that it refers to the overall farrowing of all your sows. But it was a little bit confusing when reading only the Table.
We have altered this in the table and legend.
L-127-130. Not totally clear to the reviewer. To account for a “sow” effect, there was one piglet receiving each treatment in each of the sows (ie 5 piglets controlled/sow, 2 drenching doses and 3 the sham treatments). Was this feasible, understanding that you also had to select LBW and VLBW piglets accross different sows? How was it balanced? Sorry I do not totally follow your explanation in Material and Methods. I see that this is somehow accounted for the statistical analysis, but it is not totally clear to the reviewer.
Indeed, it was not possible to always select 5 piglets per sow, as most litters only contained a few LBW or VLBW piglets. This was due to our selection criterium that was based on deviations from the litter’s mean birth weight. If we would only have selected small piglets, based on their birth weight without taking their siblings’ weight into account, the selection would have been easier. Consequently, some sows had more piglets belonging to certain treatments than others. To account for this effect, sow was included as a random factor in the model.
We have rephrased the different sections with respect to the set up.
L-139-141: All piglets were randomly allocated to one of five different treatments (by using ear tags) after they were categorised as LBW or VLBW piglets in the case of farm A and LBW piglets only in the case of farm B.
L-159-162: Within each treatment group, the number of female and male piglets (and birth weight category in the case of farm A) was equal. The litter sizes were not standardised in size. Moreover, since the treatments were allocated on the piglet level, a single sow could have 1 up to 5 piglets belonging to the different experimental groups.
L-222-225, 242-246: To account for the dependence between littermates or between measurements on the same piglet (but at a different age) random factors for sow (nested in the farrowing round) and piglet (nested in sow) were included.
L-206. One aspect that remains unclear is whether in the statistical analysis you accounted for the fact that you have repeated measures of parameters, for example for cortisol and chromogranin A. You mention that time was added as random effect. But actually what you have is a repeated measurement. How did you account for that?
We have measured e.g. body weight at different time points, which on the one hand could be considered as repeated measures but since we include the possible effect of time (= age) at those parameters we have included piglet as a random factor to account for the possible dependency of these measurements rather than run a repeated measures assay.
L-223-226, 242-246: To account for the dependence between littermates or between measurements on the same piglet (but at a different age) random factors for sow (nested in the farrowing round) and piglet (nested in sow) were included.
Figure 1. Could you indicate somehow the statistical difference that you mention in the text in the Figure too? In line 282-283 you mention that in Figure 1A there is statistical difference at p<0.001 between LBW and VLBW, but then under the Figure you indicate “mixed models p<0.05”. It is a bit confusing. Same for Figure 2 and 4, lines in text and Figure do not indicate the same statistical difference. In Figure 5, text mentions no significant effect, whereas under the Figure again it appears “linear mixed models, p<0.05”
We have altered the figures and removed the e.g. “linear mixed models p < 0.05” which was an indication of the used model and the cut-off value for significance. We agree that this was confusing to readers and was already described in the statistical analysis.
Figure 8. Maybe because of the pdf, the Chromogranin A graph is a bit cut in the manuscript I downloaded.
We have altered the figure and, based on suggestions from other reviewers, moved this part to the supplementary material.
Round 2
Reviewer 2 Report
Overall, the results indicate that farm perinatal management affects survival of low-birth weight piglets. As presented attempts to improve survival with an oral nutrient supplement failed to show beneficial responses, likely attributed to the relatively small amount of energy provided in the dose.